# How Environmental Knowledge and Green Values Affect the Relationship between Green Human Resource Management and Employees' Green Behavior: From the Perspective of Emission Reduction

Shaoying Zhu *, Yuxin Wu and Qian Shen

School of Business Administration, Guangxi University, Nanning 530004, China; zishanyimeng@163.com (Y.W.); shen1995qian@163.com (Q.S.)
* Correspondence: 19950069@gxu.edu.cn

**Abstract:** Green human resource management (GHRM) determines the green behavior practice of employees and affects the social environment and the realization of "Beautiful China" and "Green Development". In this study, to explore the impact mechanism of GHRM on employees' green behavior, employees at all levels in an enterprise were selected to be research subjects and a regulated intermediary model was established, based on social exchange theory and the individual-environment matching theory. This paper investigated the enterprise's GHRM, personal green behavior, relational psychological contract, environmental knowledge and green values. The results show that GHRM has a significant positive predictive effect on employees' green behavior, the relational psychological contract plays an intermediary role between GHRM and employees' green behavior and the intermediary role of the relational psychological contract is regulated by environmental knowledge and green values. These research results explain the relationship between GHRM and employees' green behavior and provide an important basis for enterprises to implement GHRM practice and promote employees' green behavior.

**Keywords:** green human resource management; employee green behavior; relational psychological contract; environmental knowledge; green values





## 1. Introduction

In the past few decades, irreversible climate change, widespread environmental pollution and an increasingly serious shortage of resources have constantly threatened human life. At the same time, with the rapid development of the social economy, people's pursuit of economic growth is no longer limited to "quantity", but they pay more attention to "quality". In order to achieve sustainable development, environmental impact becomes the main concern of a modern economy while ensuring the quality and quantity of economic development [1]. Enterprises have come to realize that the maintenance of their business depends on the continuous supply of natural resources. The depletion of the natural resources on which an operation depends will not only destroy the ecological sustainability but will also destroy the financial sustainability of the organization [2]. In addition, business leaders find that the triple-bottom-line strategy, considering the social, environmental and economic aspects of the enterprise, is the key to achieve a competitive advantage. "Green" began to be involved in various functional areas of the organization, including green procurement [3], green supply chain [4], green accounting [5], green innovation [6] and green human resource management (GHRM) [7].

The increasing global attention to the environment forces organizations to adopt the practice of GHRM for promoting employees' environmental behavior at work [8]. GHRM represents the link between the organization's environmental management system and human resources management system. The practice of enterprise environmental management

is directly related to human resource management. Human resources constitute the life of the organization, promote its integration with environmental management and promote the success of enterprise environmental management [9]. Today, we see that sustainable development and green organizations are receiving growing attention in management plans. As the main group for enterprises to implement the organization's green policy, the organization must promote and finally change the behavior of employees to make their behavior consistent with the organization's green goal. An increasing number of enterprises have also begun to encourage employees to take more environmental protection actions through policies related to human resource management, which plays an important role in promoting the greening of enterprises [10].

Generally, judging from the existing relevant literature, employees' green behavior is regarded as the outcome variable of GHRM [11–14]. Evangelinos et al. found that the supportive working environment characterized by green human resources practice is positively correlated with employees' willingness to create and implement environmental protection ideas [15]. Saeed et al. found that GHRM practices significantly affect employees' pro-environmental behavior and pro-environmental psychological capital plays an intermediary role in this process by investigating 347 employees working in the coal, electric power, food, chemical and pharmaceutical industries [16]. Meanwhile, most of the academic research on GHRM still focuses on enterprise strategy and operation, such as discussing how to integrate the green concept into enterprise management practice and analyzing the relationship between GHRM practice and sustainable development, organizational competitive advantage, enterprise performance and so on. For example, Yong Jong Kim et al. explored the impact of GHRM on employees' environmental behaviors and environmental performances [17], Ahmed A. Zaid pointed out that GHRM and green supply chain management practices have a positive impact on the sustainable performances of enterprises [18] and Edyta Bombiak found that there is a strong positive correlation between individual activities in GHRM and the sustainable development and practice of enterprises [19].

In summary, the current research perspective of GHRM is rarely focused on employees themselves, and the research on the relationship between GHRM and employee green behavior does not take into account the influential factor of the employee–organization relationship. Moreover, there is a lack of empirical research on the impact mechanism between GHRM and employees' green behavior, based on theory. However, the practical results of GHRM are the daily behaviors of employees that can really put the green behavior policy of the organization into practice and promote and publicize the values of the organization to improve organizational performance, thereby achieving sustainable development. Therefore, this paper argues that it is necessary to further explore the mechanism and results of GHRM on employee behavior, which is of great significance to fully understand the effectiveness of the practical measures, and to enrich the theoretical research in related fields.

According to social exchange theory, there is a social exchange relationship between employees and organizations. Employees pay attention to their efforts and rewards in this social exchange relationship. Employees' green behavior itself is a behavior out of role. Whether it will occur depends more on the judgment of the relationship between themselves and the organization. The relational psychological contract holds that employees will make efforts beyond their responsibility in order to repay the organization when they perceive that there is a long-term, stable and mutually responsible relationship between themselves and the organization. Therefore, when there is a strong relational psychological contract in the organization and GHRM is emphasized in the organization, employees may implement green behaviors according to organizational requirements.

The individual-environment matching theory holds that the matching of knowledge and values between employees and enterprises is conducive to the establishment of harmonious relations between them. At present, there is a lack of research on introducing green values into the relationship between GHRM and the relational psychological contract

and exploring the boundary conditions between GHRM and the relational psychological contract. The practice of GHRM should eventually be implemented by the employees of the enterprise. When employees have sufficient knowledge and corresponding values in this regard, they may have a higher sense of belonging to the organization and maintain an optimistic attitude towards the establishment of the relationship between themselves and the organization. Specifically, organizations will select candidates who match their own values or have relevant knowledge and provide green training to help them manage their knowledge in the GHRM practice so that they can have a deeper understanding of environmental protection. Therefore, the environmental knowledge and green values of the employees in the organization may also affect the establishment of employee relations in specific practices.

As a result, a regulatory intermediary model will first be established by analyzing the intermediary role of the relational psychological contract. Then, the possible regulatory role of environmental knowledge and green values will be discussed, based on the research on the direct relationship between GHRM and employees' green behavior. Finally, a new model of the impact mechanism of GHRM on employees' green behavior will be constructed by systematically analyzing the mechanism of GHRM on employees' green behavior. This will help us to have a deeper understanding of the impact mechanism of GHRM on employees' green behavior, improve enterprise managers' understanding of the new management concept of GHRM and provide a theoretical basis for enterprises to better guide employees to green behavior.

## 2. Conceptual Model and Literature Review

The literature on human resource management behavior shows that the human resource management practice implemented by the organization affects the overall performance of the organization by affecting employees' work behavior and attitude [20]. When employees perceive the human resource management practice implemented by the organization, they will consciously adjust their work attitude and behavior according to this standard [21]. That is to say, when an organization brings green into its human resource management policy, employees will show behaviors that resonate and comply with the organization's green policy. Cherian and Jaco also pointed out that GHRM encourages employees to take more responsible actions to protect the environment [22]. GHRM practice can bring a higher efficiency and lower costs to the organization and create a better atmosphere of employee relations, which in turn helps the organization operate in a friendly environment.

Rousseau defined a psychological contract as an individual's belief in the terms and conditions of mutual exchange agreements between employees and organizations and proposed a two-dimensional model of a psychological contract that divides a psychological contract into a transactional psychological contract and a relational psychological contract [23]. Social exchange theory shows that psychological contracts prompt employees to repay the treatment provided by their organizations [24].

HRM practices are assumed to affect psychological contracts because they are a part of the organization's obligations to employees, which employees consider as an inducement [25]. Arthur divided human resource systems into two types: control-based and commitment-based [26]. A control-based human resource system relies on forcing employees to comply with specified rules, procedures and result-based rewards to improve efficiency and directly reduce labor costs. A commitment-based human resource system has widely defined jobs, broader and general skills training, higher wages and more extensive benefits [27]. When an organization provides these incentives, employees will perceive the organization's commitment to stable and long-term employment and support for the well-being and interests of the employees and their families. These concepts will help shape employees' relational psychological contracts. In order to verify this result, UEN et al. linked the commitment-based human resource management system with the employee-level psychological contract. The research results showed that human resource

management practice has a positive impact on the relational psychological contract. GHRM can be understood as a practice of human resource management based on commitment. It is not a rigid or mandatory requirement for employees but motivates employees to adapt to the green practice of the organization, with the help of various management activities. This practice of human resource management, based on commitment, will guide employees to show the behavior expected by the organization in regards to performance, salary and participation in management, so as to finally establish a stable relationship with employees' relational psychological contract of long-term commitment. On the other hand, GHRM practice makes employees obtain economic satisfaction through green performance management and green salary management. At the same time, employees' recognition of green practice will also make them obtain social and emotional satisfaction, which is more likely to shape employees' relational psychological contract. Therefore, GHRM is expected to have a positive impact on the relational psychological contract.

Employees often perform what they think is needed by the organization, according to their psychological contract, and they are more inclined to perform their beliefs according to their own psychological contract. Therefore, a psychological contract has a positive relationship with employees' green behavior, green motivation, commitment and trust. Shore et al. pointed out that employees' behavior can be adjusted through a psychological contract. Employees measure their behavior through their responsibility to the organization and standardize their behavior [28]. In other words, a psychological contract is formed by human resource management and has an impact on employees' behavior. Employees will have their own views about the organization-based behaviors (GHRM practice), which in turn will determine their green behavior to repay the organization.

According to the psychological contract theory, employees believe that an employers' unwritten commitment to them, in terms of training, promotion and other ambiguous factors, plays a vital role in developing a good communication relationship. A psychological contract can be regarded as the belief built by employees from the human resources system [29] and a relational psychological contract represents that this belief is stable and long-term. Robinson and Morrison believe that a psychological contract is particularly important for the evaluation of employees' out-of-work behavior [30]. If a psychological contract takes effect actively, it can maximize the performance of the organization and its employees, because a psychological contract can directly affect the members' work attitude and behavior towards his or her organization. At the same time, a large number of studies show that the impact of a relational psychological contract on employee behavior is significantly stronger than a transactional psychological contract. Under the relational psychological contract, employees will easily accept and recognize the GHRM practice implemented by the organization and show the behavior the organization wants. In other words, employees' perception of the relational psychological contract mediates the relationship between GHRM practice and employees' green behavior. Therefore, a relational psychological contract is considered as the intermediary relationship between GHRM practice and employees' green behavior.

Environmental knowledge refers to the knowledge and awareness of environmental problems and their solutions [31]. The individual-environment matching theory holds that the matching of knowledge and values between employees and enterprises is conducive to the establishment of harmonious relations between the two sides. Kristof pointed out that individual-environment matching improves employees' satisfaction and commitment to the organization. Zhao Huijuan and long Lirong also found that individual-environment matching can significantly and positively predict employees' emotional commitment [32]. Therefore, employees' knowledge and values will have an impact on their decision-making and intention. They will judge the organization according to their own knowledge and values in the process of choosing the organization or in the working process. The judgment result will determine the relationship in different organizations.

Barr pointed out that when employees' knowledge of waste management, environmental management systems and enterprise green policy increases, they will enhance their

recognition of organizational green management, which may increase their green behavior in the workplace [33]. In other words, when individuals have a better understanding of environmental problems, processes and solutions, their awareness and understanding of their role in protecting the environment will be correspondingly improved so that individuals will have a higher recognition of the GHRM practice implemented by the organization, which will increase their sense of integration and responsibility to the organization. It is more likely to establish a relationship of a relational psychological contract with the organization. This lasting and stable relationship will further promote employees to perform more green behaviors at work.

The existing values literature emphasizes the importance of personal values in explaining personal attitudes and behaviors [34]. Maslach et al. found that the higher the degree of matching of values between individuals and organizations, the easier their needs will be met [35]. Supply-value fit theory also holds that when employees' personal values are consistent, it will have a positive impact on employees' work attitude and behavior. However, it is inevitable that there are some contradictory values between individuals and organizations. When personal values and organizational values are combined to form a common ideology, employees' commitment to the organization will be strengthened [36]. Then, employees are easier to form a long-term and stable relationship with the organization, that is, a relational psychological contract. The closer an individual connects with the organization through values and sense of identity, the more likely employees are to commit to achieving organizational goals [37]. Poortinga et al. pointed out that values have no strong direct impact on behavior, and the relationship between general values and behavior is mediated by other factors, such as behavior-specific beliefs or personal norms [38]. That is to say, if an organization provides an environment conducive to employees' values, makes employees' green values consistent with the organization's and forms a strong sense of belonging to the organization, employees will be more likely to show the green behavior in their work. On the contrary, if the employee's values are inconsistent with the organization's values or the organization does not provide an environment matching the employee's needs, the employee's sense of belonging to the organization will be reduced accordingly, and the possibility of green behavior will be reduced. In addition, a relational psychological contract can also be regarded as the result of employees' judgment on organizational values. Rupp et al. pointed out that employees will make clear judgments on the organization's social responsibility policies and behaviors that determine whether employees' psychological needs are met [39]. Under the relational psychological contract, the fit between employees' green values and organizational values will increase employees' sense of belonging to the organization and the recognition of the green practice implemented by the organization so that they are more likely to establish a long-term cooperative relationship.

To explore the impact mechanism of GHRM on employees' green behavior, this study assumes that a relational psychological contract plays a mediating effect between GHRM and employees' green behavior, and environmental knowledge and green values play a moderating effect on the first half of the mediating effect. The research model is shown in Figure 1.

GHRM may affect employees' green behavior through the following aspects. Firstly, enhancing employees' awareness and understanding of green behaviors by communicating the organization's preference for green behaviors in the recruitment process and taking personal environmental values into the consideration in the employee selection process [7], which ensures the new employees' quicker adaptation to the green management of organization and more green behaviors. Secondly, work and work design, which can meet environmental requirements, and green training with their aims at improving employees' knowledge, skills and abilities are the key processes to encourage employees to have green behavior [40]. Tseng et al. mentioned that organizations can effectively increase employees' attention and motivation to environmental protection behavior by encouraging employees to understand the environment and providing regular and frequent training on the environ-

mental management system during the design of work and the working environment [41]. In addition, employees' cognition of the reasons the organization adopts some human resource management practices determines the effectiveness of human resource management practices on employees' work behavior [42]. A set of formal and public GHRM practices and policies that voices the company's commitment to green management may result in an employee's compliance with the company's green policies [12]. Moreover, when the organization introduces the reward of environment-friendly performance, employees will actively participate in the organization's green practices and facilitate their green activities. The positive incentive of giving praises or rewards to employees for their good green performance can effectively increase employees' recognition of GHRM practice and maintain or increase green behavior in their future work. Finally, GHRM provides employees with opportunities to participate in management and encourages them to actively contribute to the organization's green practices. In the process of participating in management, employees will have a greater sense of support for green practice and ensure their attitude is consistent with their behaviors and organizational goals. In other words, employees will have more green behaviors.

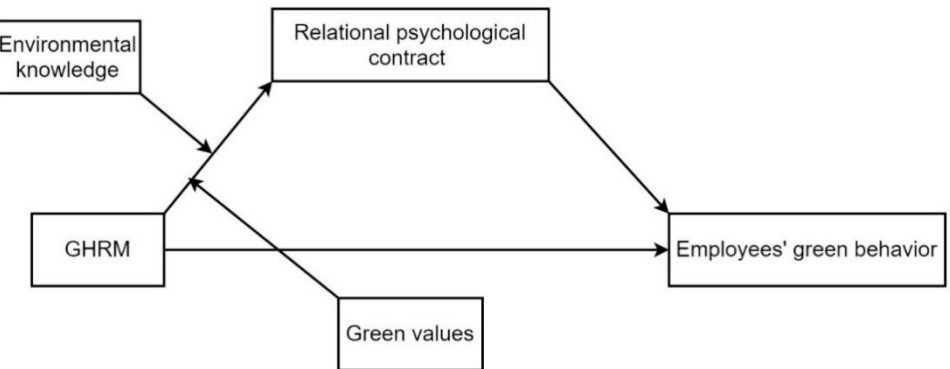

**Figure 1.** Theoretical research model (a regulated intermediary model).

According to social exchange theory, employees have a social exchange relationship with their organizations, which makes them pay attention to their efforts and the rewards of the social exchange relationship. The performance of a psychological contract means that the organization has fulfilled its obligations to employees as expected, which has a positive impact on employees' attitude, behavior and performance. Hui et al. found that a relational psychological contract has a positive impact on employees' organizational citizenship behavior [43]. Rosen et al. provided empirical evidence that employees' views on political and procedural justice affect the psychological contract, work attitude and situational performance, all of which will affect work behavior in terms of organizational citizenship behavior [44]. Kiazad et al. considered that when employees believe that the organization meets their expectations and emphasize the establishment of long-term relationships, they will show a strong willingness to organizational citizenship behavior [45]. That is to say, the organization will provide employees with economic, social and emotional satisfaction for the relational psychological contract. At the same time, a long-term, stable and future-oriented relationship has been formed between the organization and employees, and both sides are responsible for each other's development. Therefore, employees hold a more positive attitude towards the relationship between the two sides and have a strong sense of belonging to the organization that will form a behavioral motivation to drive employees to devote their work efforts, so as to improve employees' recognition of the green practice, show more green behaviors and promote the success of the GHRM practice.

Although there is evidence that GHRM can lead to green behavior, according to its definition, GHRM leaves employees considerable freedom. The interaction between environmental knowledge and GHRM may strengthen employees' willingness to cooperate with the organization, which also enhances the establishment of a relational psychologi-

cal contract between the organization and employees, thus increasing the generation of employees' green behavior [46].When employees are aware of environmental problems, the matching between the GHRM policy and its judgment will make employees have a better understanding of the organization's green policy. The matching between their own knowledge and the organization's policy will make it easier for employees to establish a long-term cooperative relationship with the organization. Therefore, the more environmental knowledge employees have, the more likely they are to establish a relational psychological contract. That is, employees' environmental knowledge can adjust the impact of GHRM on the relational psychological contract.

## 3. Samples and Measurements

### 3.1. Research Samples

This study collected relevant data by a questionnaire survey. Considering that the concept of the green management of enterprises in economically developed areas is more advanced than that in economically underdeveloped areas, the survey selected enterprises from the 10 provinces (Municipality and Autonomous) of Beijing, Shanghai, Guangdong, Hubei, Hunan, Sichuan, Zhejiang, Shandong, Henan and Chongqing. The respondents were ordinary employees and junior, middle and senior managers of enterprises. A total of 313 questionnaires were sent out, and 271 valid questionnaires were obtained after eliminating the invalid questionnaires, with a total effective recovery rate of 86.6%. The sample characteristics are shown in Table 1.

**Table 1.** Sample analysis (*N* = 271).

|  |  | **Frequency** | **Percent** |
|---|---|---|---|
| Gender | Men | 159 | 58.67% |
|  | Women | 112 | 41.33% |
| Age | 18–30 years | 41 | 15.13% |
|  | 30–40 years | 110 | 40.59% |
|  | 40–50 years | 94 | 34.69% |
|  | Over 50 years old | 26 | 9.59% |
| Education | A high school degree or less | 25 | 9.23% |
|  | Junior college | 54 | 19.93% |
|  | Bachelor | 104 | 38.38% |
|  | Master degree or above | 88 | 32.47% |
| The nature of enterprises | State-owned enterprise | 54 | 19.93% |
|  | Private enterprise | 116 | 42.80% |
|  | Administration | 27 | 9.96% |
|  | Others | 74 | 27.31% |
| Years of working | 1–3 years | 22 | 8.12% |
|  | 4–6 years | 17 | 6.27% |
|  | 7–9 years | 28 | 10.33% |
|  | Over 10 years | 204 | 75.28% |
| The level of work | Ordinary employee | 79 | 29.15% |
|  | Junior manager | 59 | 21.77% |
|  | Middle manager | 71 | 26.20% |
|  | Senior manager | 62 | 22.88% |

Through the descriptive analysis of the demographic data in this study, it can be found that men account for 58.67% and women accounted for 41.33% in the survey group. In terms of age, people aged 30–40 and 40–50 accounted for the vast majority, with percentages of 40.59% and 34.69%, respectively. In terms of education level, the proportion of undergraduate and master's degrees were 38.38% and 32.47%, respectively, which shows that the sampled groups in this study were well educated. In terms of enterprise nature, private enterprises accounted for the highest proportion, reaching 42.80%. In terms of

working years, people with over 10 years of working experience accounted for the vast majority, with a proportion of 75.28%. In terms of hierarchy, all of the four levels of work shared almost the same percentage, accounting for 29.15%, 21.77%, 26.20% and 22.88%.

*3.2. Variable Measurement*

In this paper, GHRM, employees' green behavior, relational psychological contract, environmental knowledge and green values were taken as variables. Well-developed scales by foreign experts and scholars were employed in this study. A six-point Likert scale method was adopted (1 = "strongly disagree", 2 = "disagree", 3 = "partially disagree", 4 = "partially agree", 5 = "agree", 6 = "strongly agree"). The surveyed group selected the options according to their enterprises' actual situation and their own behaviors.

A scale developed by Tang et al. (2018) was chosen to measure GHRM with 18 items, such as "our company recruits employees with environmental awareness", whose Cronbach's $\alpha$ was 0.987. A scale developed by Robertson and Barling (2013) was used to measure an employee's green behavior including six items, such as "I print on both sides as much as possible", whose Cronbach's $\alpha$ was 0.885. A scale developed by Millward and Hopkins (1998) was used to measure the relational psychological contract with six items, such as "I want to promote my personal growth in this organization", whose Cronbach's $\alpha$ was 0.955. A scale developed by Gatersleben, Steg and Vlek (2002) was used to measure environmental knowledge with nine items, such as "I have a better understanding about the environmental problems", whose Cronbach's $\alpha$ was 0.943. A scale developed by Steg et al.(2006) was introduced to measure green values with eight items, such as "I will be better if I save energies", whose Cronbach's $\alpha$ was 0.947.

## 4. Results

*4.1. Common Method Deviation Inspection*

The questionnaires in this study were completed anonymously and independently by one person, so confirmatory factor analysis (CFA) was used to test the possible common method deviation. The CFA test showed that the fitting index of the single factor model (x2/DF = 13.89, RMSEA = 0.15, CFI = 0.525, TLI = 0.506) was not ideal, so it can be explained that there is no obvious common method deviation.

*4.2. Descriptive Statistics and Correlation Analysis*

The means, standard deviations and correlation coefficients of the five variables in the study are shown in Table 2. It can be seen from the table that GHRM was significantly positively correlated with employees' green behavior (r = 0.536, *p* < 0.01) and relational psychological contract (r = 0.542, *p* < 0.01). The relational psychological contract was significantly positively correlated with employees' green behavior (r = 0.734, *p* < 0.01). Environmental knowledge (r = 0.714, *p* < 0.01) and green values (r = 0.705, *p* < 0.01) were significantly positively correlated with the relational psychological contract. This shows that the hypotheses proposed in this study were preliminarily affirmed, which provides a basis for further analysis.

**Table 2.** Descriptive statistical analysis results.

| Variables (*N* = 271) | M | SD | 1 | 2 | 4 | 5 |
|---|---|---|---|---|---|---|
| 1.GHRM | 4.59 | 1.46 | 1 | | | |
| 2. Employees' Green Behavior | 4.96 | 0.94 | 0.536 ** | 1 | | |
| 3. Relational Psychological Contract | 5.02 | 0.87 | 0.542 ** | 0.734 ** | | |
| 4. Environmental Knowledge | 5.00 | 0.87 | 0.439 ** | 0.714 ** | 1 | |
| 5. Green Values | 5.17 | 0.83 | 0.347 ** | 0.705 ** | 0.771 ** | 1 |

Note: ** represents *p* < 0.01

### 4.3. Test of the Mediating Effect of the Relational Psychological Contract

This study used an SPSS macro process compiled by Hayes (2012) to test the direct effect and intermediary effect of the hypothesis. Here, Model 4 (simple mediation model) was selected to test the results, and the results are shown in Tables 3 and 4. The test results showed that the direct effect of GHRM on employees' green behavior was significant ($\beta = 0.345$, $t = 10.401$, $p < 0.001$), and the direct effect was still significant after adding the intermediary variable of the relational psychological contract ($\beta = 0.125$, $t = 4.058$, $p < 0.001$). At the same time, GHRM had a significant positive predictive effect on the relational psychological contract ($\beta = 0.322$, $t = 10.590$, $p < 0.001$), and the positive predictive effect of the relational psychological contract on employees' green behavior was also significant ($\beta = 0.684$, $t = 13.131$, $p < 0.001$). In addition, the bootstrap 95% confidence intervals of the direct effect of GHRM on employees' green behavior and the intermediary effect of the relational psychological contract were (0.011, 0.440) and (0.146, 0.303), respectively, and the upper and lower limits of the confidence intervals did not contain 0, which indicated that GHRM not only directly predicted employees' green behavior but also briefly predicted employees' green behavior through the relational psychological contract. At the same time, the direct effect and intermediary effect accounted for 36.23% and 63.77% of the total effect, respectively. Therefore, the green behavior of employees involved in this study had an intermediary role, and the relational psychological contract played a partial intermediary role.

**Table 3.** Intermediary model test of the relational psychological contract.

| Result Variable | Predictor Variable | R | $R^2$ | F | β | t |
|---|---|---|---|---|---|---|
| Employees' Green Behavior | GHRM | 0.536 | 0.287 | 108.187 *** | 0.345 | 10.401 |
| Relational Psychological Contract | GHRM | 0.542 | 0.294 | 112.154 *** | 0.322 | 10.590 |
| Employees' Green Behavior | GHRM Relational Psychological Contract | 0.752 | 0.566 | 174.785 *** | 0.125 0.684 | 4.058 13.131 |

Note: *** represents $p < 0.001$.

**Table 4.** Decomposition table of total effect, direct effect and intermediary effect.

| | Effect Value | Boot SE | BootLLCI | BootULCI | Relative Effect Value |
|---|---|---|---|---|---|
| Total Effect | 0.345 | 0.048 | 0.251 | 0.440 | |
| Direct Effect | 0.125 | 0.061 | 0.011 | 0.249 | 36.23% |
| Intermediary Effect of Relational Psychological Contract | 0.220 | 0.041 | 0.146 | 0.303 | 63.77% |

### 4.4. Test on the Regulatory Effect of Environmental Knowledge and Green Values

In terms of testing the mediation model with regulation, this study tested the hypotheses proposed in the study with the help of the SPSS model 7 macro process (variables regulate the first half of the mediation model) compiled by Hayes (2012). The results are shown in Tables 5 and 6. After involving environmental knowledge into the mediation model, GHRM significantly predicted the relational psychological contract ($\beta = 0.162$, $t = 6.648$, $p < 0.001$), and environmental knowledge also significantly predicted the relational psychological contract ($\beta = 0.679$, $t = 16.137$, $p < 0.001$). In addition, the product of GHRM and environmental knowledge also significantly predicted the relational psychological contract ($\beta = 0.054$, $t = 2.621$, $p < 0.01$), which showed that environmental knowledge can regulate the predictive effect of GHRM on the relational psychological contract.

**Table 5.** Mediation model test with adjustment.

| Result Variable | Predictor Variable | R | R$^2$ | F | β | t |
|---|---|---|---|---|---|---|
| Relational Psychological Contract | GHRM | 0.803 | 0.644 | 161.185 *** | 0.162 | 6.648 |
| | Environmental knowledge | | | | 0.679 | 16.137 |
| | GHRM×Environmental Knowledge | | | | 0.054 | 2.621 |
| Relational Psychological Contract | GHRM | 0.800 | 0.641 | 158.599 *** | 0.202 | 8.682 |
| | Green Values | | | | 0.689 | 15.968 |
| | GHRM × Green Values | | | | 0.077 | 3.414 |

Note: *** represents $p < 0.001$.

**Table 6.** Intermediary effect of the relational psychological contract under different adjustment levels of environmental knowledge and green values.

| Moderating Variables | Regulated Level | Effect Value | Boot SE | BootLLCI | BootULCI |
|---|---|---|---|---|---|
| Environmental Knowledge | M − 1SD | 0.0786 | 0.0300 | 0.0201 | 0.1389 |
| | M | 0.1108 | 0.0283 | 0.0631 | 0.1728 |
| | M + 1SD | 0.1430 | 0.0414 | 0.0776 | 0.2370 |
| Green Values | M − 1SD | 0.0947 | 0.0344 | 0.0275 | 0.1603 |
| | M | 0.1383 | 0.0288 | 0.0859 | 0.1972 |
| | M + 1SD | 0.1818 | 0.0427 | 0.1083 | 0.2733 |

In order to observe the regulatory effect of environmental knowledge more clearly, this study further made a simple effect analysis diagram of GHRM on the relational psychological contract (Figure 2). It can be seen from the figure that when environmental knowledge was at a low level (m − 1sd), GHRM significantly positively predicted the relational psychological contract (simple slope = 0.028, $t$ = 4.103, $p < 0.001$). When environmental knowledge was at a high level (M + 1sd), GHRM also had a significant positive predictive effect on the relational psychological contract, but its predictive effect was higher than the former (simple slope = 0.032, $t$ = 6.461, $p < 0.001$), which indicated that with the improvement of employees' environmental knowledge level, the predictive effect of GHRM on the relational psychological contract was gradually increasing. In addition, the intermediary effect value and bootstrap 95% confidence interval between GHRM and the relational psychological contract under different environmental knowledge levels (m − 1sd; m; m + 1sd) are shown in Table 6. At different environmental knowledge levels, the upper and lower limits of the confidence interval did not contain 0, which was consistent with the results of the simple effect analysis. At the three levels of environmental knowledge, the mediating effect of the relational psychological contract between GHRM and employees' green behavior was gradually strengthened In other words, with the improvement of employees' environmental knowledge, it was easier for GHRM to promote employees' green behavior by strengthening employees' relational psychological contract.

In addition, the SPSS model 7 macro process compiled by Hayes (2012) was also used to test the regulatory effect of green values in the first half of the mediation effect. As shown in Table 5, after putting the green values into the intermediary model, GHRM had a significant positive predictive effect on the relational psychological contract (β = 0.202, $t$ = 8.682, $p < 0.001$). Green values also significantly positively predicted the relational psychological contract (β = 0.689, $t$ = 15.968, $p < 0.001$). The interaction between GHRM and green values significantly positively predicted the relational psychological contract (β = 0.077, $t$ = 3.414, $p < 0.001$). The results showed that green values played a regulatory role between GHRM and the relational psychological contract.

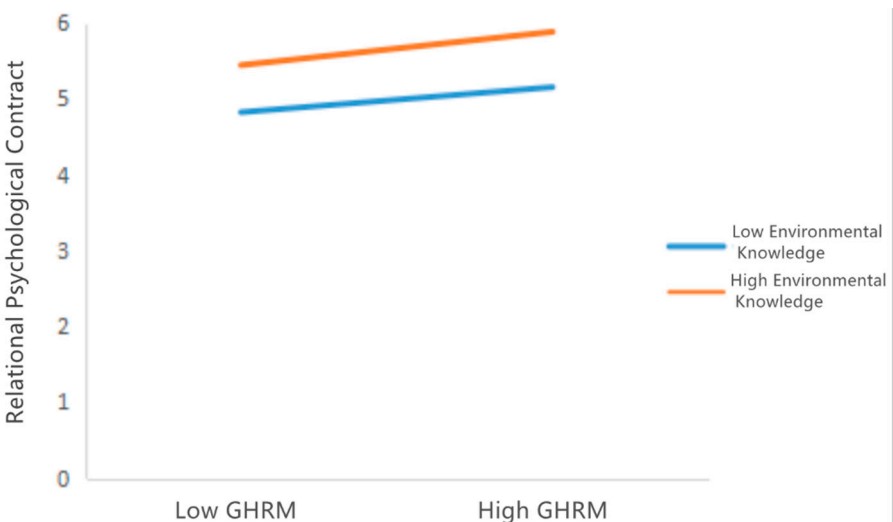

**Figure 2.** Mediating effects of environmental knowledge between GHRM and relational psychological contracts.

Furthermore, from the simple effect analysis chart (Figure 3), when the green values were at a reduced level (m − 1sd), GHRM significantly positively predicted the relational psychological contract (simple slope = 0.029, *t* = 4.838, *p* < 0.001). When the green values were at the high level (M + 1sd), the positive predictive effect of GHRM on the relational psychological contract was enhanced (simple slope = 0.031, *t* = 8.572, *p* < 0.001), This showed that with the improvement of employees' green values, the predictive effect of GHRM on the relational psychological contract was also increasing. In addition, the intermediary effect value and bootstrap 95% confidence interval between GHRM and the relational psychological contract under different green value levels (m − 1sd; m; m + 1sd) are shown in Table 6. At different green value levels, the upper and lower limits of the confidence interval did not contain 0, which was consistent with the results of the simple effect analysis. Although the mediating effect of the relational psychological contract was significant at different levels of green values, with the continuous improvement of the level of green values, the mediating effect of the relational psychological contract between GHRM and the employees' green behavior was also increasing.

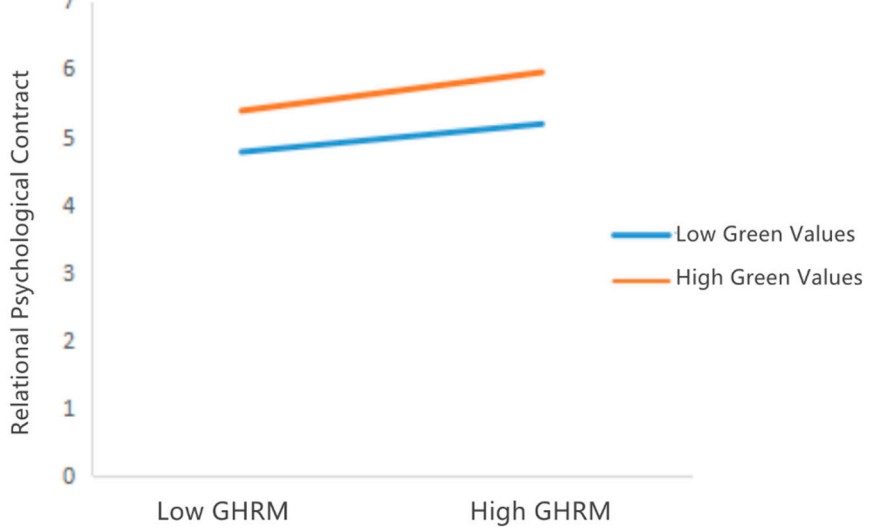

**Figure 3.** Moderating effect of green values between GHRM and the relational psychological contract.

In addition to the separate analysis of the regulatory role of environmental knowledge and green values, this study further used the SPSS Model 9 macro process compiled

by Hayes (2012). The results showed that when environmental knowledge and green values were at a low level, the mediating effect of the relational psychological contract was significant, with a value of 0.0711, and the bootstrap 95% confidence interval was (0.0116, 0.1332). Similarly, when environmental knowledge and green values were at a high level, the mediating effect of the relational psychological contract was also significant, but the significant effect was significantly higher than that at the low level. The mediating effect value was 0.1434, and the bootstrap 95% confidence interval was (0.0807, 0.2358) The results were consistent with the previous analysis results. With the improvement of employees' environmental knowledge and green values, the mediating role of the relational psychological contract was increasing.

## 5. Discussion and Implications

Through a regulated intermediary model, this study discusses the impact mechanism and boundary conditions of GHRM on employees' green behavior, which provides a certain reference value for the related research of GHRM, employees' green behavior, the relational psychological contract, environmental knowledge and green values.

Firstly, this study discusses the impact mechanism of GHRM on employees' green behavior in China and finds that GHRM can effectively increase employees' green behavior, which is helpful to further understand the impact mechanism of GHRM.

Secondly, based on social exchange theory, this paper puts forward the intermediary role of the relational psychological contract between them, which further broadens the intermediary mechanism of GHRM on employees' green behavior.

Finally, under the guidance of the individual-environment matching theory, this paper tests the regulatory effect of environmental knowledge and green values between GHRM and the relational psychological contract, which provides a new perspective for the related research of GHRM.

In addition, GHRM can effectively promote the establishment of employees' green behavior and relational psychological contract. Therefore, it is necessary for enterprises to emphasize the organization's green tendency in the process of recruitment, training and performance management and take specific measures to form a green atmosphere within the enterprise to have a subtle impact on employees. In addition, enterprises can also incorporate environmental issues into the job description and design, and green standards should also be included in recruitment information. Meanwhile, enterprises should pay attention to the establishment of the relational psychological contract with employees. The relational psychological contract can significantly increase employees' supportive behavior to the enterprise, which will contribute to the implementation of various policies and practices. Therefore, enterprises should provide opportunities for learning, enlarge employees' knowledge of the environment and foster employees' green values, which not only contributes to the successful implementation of GHRM but also promotes the establishment of a long-term cooperative relational psychological contract.

Our study also has significant implications for practice.

Firstly, this study explores the mechanism of GHRM's influence on employees' green behavior. Employees' green behavior is a key factor to improve the green performance of the organization. Through the implementation of GHRM practices, enterprises can promote employees' acceptance and recognition of the organization's GHRM practices from the aspects of recruitment and training and encourage employees to produce more green behavior through performance management, compensation management and employee participation. This can make more enterprises realize the importance of GHRM practice to guide employees' green behavior and make them pay enough attention to GHRM and actively implement GHRM practice.

Secondly, the research on the role of the relational psychological contract proves that a good relationship between employees and organizations will have a positive impact on their work behavior. Therefore, enterprises should stress the relationship between employees, meet their economic needs with sufficient communication and training and ultimately

promote the establishment of a long-term and stable relationship-based psychological contract between employees and organizations.

Finally, through the research on the moderating effect of environmental knowledge and employees' green values, it was found that the fit between employees' values and the organization can enhance employees' sense of belonging to the organization and constantly improve organizational learning, thus exerting a positive impact on employees' behavior. Therefore, enterprises should help employees build up their green values and strengthen the publicity of environmental knowledge, which not only helps to establish a good relationship between employees and enterprises but also further influences the green behaviors of employees.

### 6. Limitations, Future Research and Conclusions

In this study, we established and verified a model. The results of the study are as follows:

Firstly, GHRM can have a positive impact on employees' green behavior. Employees' participation in enterprise green management can effectively enhance their sense of responsibility for the organizational practice and enable employees to have more green behaviors to support the enterprise GHRM practice.

Secondly, as a long-term and stable relationship, a relational psychological contract exists between GHRM and employees' green behavior. Under the relational psychological contract, employees are more likely to accept and recognize the GHRM practice, so as to follow the requirements of the GHRM practice and improve employees' willingness to carry out green behavior. Thirdly, environmental knowledge and green values play a positive regulatory role between GHRM and the relational psychological contract. With the continuous improvement of employees' environmental knowledge level, their understanding and recognition of organizational GHRM practice will continue to improve, and they will be more willing to adopt green behavior in their work.

GHRM practices may vary between different companies or sectors. Limited by time and resources, the surveyed samples of this study were restricted to EMBA and MBA students and some employees of central enterprises. Their engagement in scattered sectors or regions made the study less targeted. Future research could focus on a specific sector or region.

Meanwhile, any human resource management practice may take a long time to exert the greatest impact on the results of employees' work behavior. The data of this study were collected at one time point. Therefore, this research design may not fully explore the effect of human resource management. In order to solve this limitation, future research can consider longitudinal research to investigate the impact of GHRM on employees' green behavior.

From the perspective of incentive human resource management, a set of human resource management practices may lead to the results of a variety of employees' workplace behavior. Therefore, the effect of GHRM may go beyond the green behavior of employees.

However, the existing GHRM literature only conceptualizes the relationship between GHRM and employees' or organizations' green results. The impact of GHRM on non-green work attitude and behavior has been ignored. Therefore, the author believes that the impact of GHRM on employees' non-green attitudes and behaviors can be studied in future research.

**Author Contributions:** Conceptualization, S.Z.; methodology, S.Z. and Y.W.; software, S.Z.; investigation, Y.W. and Q.S.; data curation, Y.W. and Q.S.; writing—original draft preparation, S.Z.; writing—review and editing, Y.W. All authors have read and agreed to the published version of the manuscript.

**Funding:** This research received no external funding.

**Institutional Review Board Statement:** The study did not involve humans or animals.

**Informed Consent Statement:** The study did not involve humans.

**Conflicts of Interest:** We declare that we have no financial and personal relationship with other people or organizations that can inappropriately influence our work, there is no professional or other personal interest of any nature or kind in any product, service and company that could be constructed as influencing the position presented in, or the review of, the manuscript entitled.

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
