# Peer review of "How Environmental Knowledge and Green Values Affect the Relationship between Green Human Resource Management and Employees’ Green Behavior: From the Perspective of Emission Reduction"

_processes, doi:10.3390/pr10010038_

Round 1

Reviewer 1 Report

Dear authors, thank you for the opportunity to reads your paper. I find it interesting and quite relevant.  Several issues are suggested, as I believe they can help your paper be even better:

  • Please make a grammar check – for instance you have a mistake in your title – Emmisom instead of emission.
  • In the abstract instead of providing details about your sample, rather explain shortly the reasoning for the study and which relations you analyze in the paper.
  • In your introduction you state “However, since green human resource management is a relatively new concept, most of the existing relevant literature (eg. cherian & Jacob, 2012; daily & Huang, 2001; Jabbour, 2011; Jackson & SEO, 2010; Renwick et al, 2013“ – but these are all quite old references. This is the filed that has grown significantly in the last decade, and many new research existing – so please renew the sources.
  • The introduction should more clearly and deeply provide 1) what we currently know, 2) what we do not know and 3) how your paper contributes to closing this gap. Reason that “the domestic research on green human resource management is still in its infancy“ is not a good enough reason for your study. Moreover, it raises the question of contribution of your paper for general public.
  • Explain the reasoning for choosing all of the study variables in your introduction. Currently it seems there is no reasoning provided for using regulatory role of environmental knowledge and green values as a variable.
  • When first mentioning GHRM or HRM, then please use this abbreviation throughout the rest of the paper.
  • Please provide sample selection criteria and provide information about the sample characteristics.
  • I miss a Discussion section where main results would be discussed and presented in the light of previous similar research.
  • There is no need for subsections in the conclusion part (which should actually be named just Conclusion). Especially as you would have a Discussion section where main result would be presented. So in the conclusion you would only present theoretical and practical implications as well as research limitations and possible future research/open questions.
  • Newer references should be used.

Author Response

Please check the attachment, thank you!  

Reviewer 2 Report

Even though the proposed subject seems interesting, there is much work needed before the paper is published.

Authors should enrich their study about the subject in order to understand current challenges and issues that seem to be highly – appreciated by the academic community. Most of existing bibliography is not recent, which affects both literature review and research questions posed. Out of 34 references used only one is originated in 2015, two references are originated in 2016, one originated in 2017 and one is originated in 2018.

Research questions posed are not directly linked with existing research (coming from bibliography) and their significance / merit is not proved.

Methodology is not adequately presented. Why the sample is the most appropriate? Which are the demographic characteristics of the sample? For how long have the questionnaire been distributed? Has the proposed questionnaire ever been tested is any other study?  

Problems exist with table presenting results (see table 3.4).

Conclusions need a further discussion about results, how research questions are answered, the significance of these answers and convergencies / divergencies of results with existing studies.

Author Response

Please check the attachment, thank you!  

Round 2

Reviewer 1 Report

Dear authors, thank you for incorporating reviewers' comments. The paper seems much advanced. One minor thing still remains - Discussion and Conclusion need to be separate sections, where Discussion section will  present main results, while Conclusion section would present theoretical and practical implications as well as research limitations and possible future research/open questions.

Reviewer 2 Report

Even though more refferences have been added, these are not part of literature review but part of the paper as a whole. As a result, it is not clear which is existing research and how the proposed paper contributes to recent research developments. 

Moreover, limitations or research would benefit the whole research.

Round 3

Reviewer 2 Report

Reviewer recognizes the work and improvements made from authors. Best wishes for the new academic year.